

# Pay attention and you won't lose it: a deep learning approach to sequence imputation

Ilia Sucholutsky[1], Apurva Narayan[2,3], Matthias Schonlau[1] and Sebastian Fischmeister[2]

[1] Department of Statistics and Actuarial Science, University of Waterloo, Waterloo, Ontario, Canada
[2] Department of Electrical and Computer Engineering, University of Waterloo, Waterloo, Ontario, Canada
[3] Department of Computer Science, University of British Columbia, Kelowna, British Columbia, Canada

## ABSTRACT

In most areas of machine learning, it is assumed that data quality is fairly consistent between training and inference. Unfortunately, in real systems, data are plagued by noise, loss, and various other quality reducing factors. While a number of deep learning algorithms solve end-stage problems of prediction and classification, very few aim to solve the intermediate problems of data pre-processing, cleaning, and restoration. Long Short-Term Memory (LSTM) networks have previously been proposed as a solution for data restoration, but they suffer from a major bottleneck: a large number of sequential operations. We propose using attention mechanisms to entirely replace the recurrent components of these data-restoration networks. We demonstrate that such an approach leads to reduced model sizes by as many as two orders of magnitude, a 2-fold to 4-fold reduction in training times, and 95% accuracy for automotive data restoration. We also show in a case study that this approach improves the performance of downstream algorithms reliant on clean data.

## INTRODUCTION

The growth of the Internet of Things has led to a sharp increase in the number of real-world datasets being collected, processed, and analyzed in a decentralized way. Unfortunately such data are rarely as clean as in lab conditions and are often plagued by noise, loss, and various other quality reducing factors. However, in many areas of machine learning, algorithms are still designed with the assumption that data at time of inference is of the same quality as training data. As a result, when algorithms are trained on clean, pre-processed samples, they may fail to function as desired when performing inference on raw data. In particular, data loss is a serious issue for many algorithms. This is especially important for safety-critical systems like cars, where a failure can have very serious consequences for users. Anomaly detection, and other end-stage algorithms, need to have very high performance in such systems, but lost data may be a large factor in preventing this. As a result, for systems that

Corresponding author
Ilia Sucholutsky,
isucholu@uwaterloo.ca

analyze streams of data, a method is needed to quickly impute the missing values as closely as possible to what the true values would have been.

Deep learning techniques have been proven to be effective at learning temporal structure of data (*LeCun, Bengio & Hinton, 2015*). However, while a number of deep learning algorithms have been successfully shown to solve numerous end-stage problems like prediction and classification (*Glorot, Bordes & Bengio, 2011*; *LeCun, Bengio & Hinton, 2015*; *Abadi et al., 2016*), very few attempts have been made to use them for solving the intermediate problems of data pre-processing (*Kotsiantis, Kanellopoulos & Pintelas, 2006*; *García, Luengo & Herrera, 2015*), cleaning (*Kotsiantis, Kanellopoulos & Pintelas, 2006*; *García, Luengo & Herrera, 2015*), and restoration (*Efron, 1994*; *Lakshminarayan et al., 1996*), even though from a machine learning perspective these end-stage and intermediate problems can be very similar. Long Short-Term Memory (LSTM) networks have previously been proposed as a solution to these intermediate problems (*Zhou & Huang, 2017*; *Sucholutsky et al., 2019*), but they suffer from major bottlenecks like requiring large numbers of sequential operations that cannot be parallelized. Recently, Transformer (*Vaswani et al., 2017*), a novel encoder–decoder model that heavily uses attention mechanisms (*Luong, Pham & Manning, 2015*), was proposed as a replacement for encoder–decoder models that use LSTM or convolutional layers, and was shown to achieve state-of-the-art translation results with orders of magnitude fewer parameters than existing models. Inspired by this impressive result, we propose using a version of Transformer with modified hyper-parameters, that we will refer to in this paper as "Restorer". Restorer solves the intermediate problem of restoring missing elements in sequences of discrete data while entirely replacing the recurrent components of existing solutions. We demonstrate that such an approach leads to reduced model sizes, faster training times, and higher-quality reconstruction when compared to what we will refer to as "the LSTM model" described by *Sucholutsky et al. (2019)*.

Our contributions are as follows:

- We implement and empirically validate the neighbourhood-restriction on Transformer that was theoretically proposed by *Vaswani et al. (2017)*.
- We propose and create Restorer, a Transformer-based model tailored for imputing sequential data.
- We reproduce the LSTM model and restoration results from *Sucholutsky et al. (2019)* in Keras
- We match the state-of-the-art (SOTA) accuracy in restoring lost automotive CAN data with a model that has just 0.6% of the number of parameters of the previous SOTA model
- We improve the SOTA accuracy in restoring lost automotive CAN data by up to 7% while requiring 2–4 times less training time than the previous SOTA model.
- We demonstrate Restorer's positive impact on downstream algorithm performance in a case study where it outperforms both the benchmark and LSTM models.

The rest of this paper is divided into four sections. 'Background' will discuss existing work. 'Restorer' will describe the Restorer and its components. 'Experiments' will detail

the experimental results. Finally, 'Conclusion' will discuss some conclusions and future steps.

## BACKGROUND

### Sequence modelling with deep learning

The most popular deep learning architecture for sequence modelling is Recurrent Neural Networks (RNNs), a type of neural network with an internal feedback mechanism that can be used as a form of memory (*Williams & Zipser, 1989*). Probably the most successful extension of RNNs are LSTMs, which increase the flexibility of the internal feedback mechanism (*Gers & Schmidhuber, 2001*). More recently, LSTM-based encoder–decoder models like Seq2Seq were shown to improve the state-of-the-art in sequence modelling and the addition of attention mechanisms was shown to further increase their performance (*Sutskever, Vinyals & Le, 2014*). In such models, the encoder performs an embedding of the entire input sequence before the decoder begins to use this embedding as input when generating an output sequence.

There has recently been much discussion about whether recurrent models provide any intrinsic advantage over feed-forward models (*Miller & Hardt, 2018*). For example, *Vaswani et al. (2017)* recently proposed that an encoder–decoder model they call Transformer, can be built that entirely eschews recurrent components. Transformer makes use of several attention mechanisms to form an architecture that significantly outperforms the state-of-the-art on machine translation tasks without resorting to recurrent layers (*Vaswani et al., 2017*). While Transformer was demonstrated to work specifically with the task of machine translation, its impressive performance suggests that it is not unreasonable to expect it can be adapted to work with other sequence modelling tasks.

### Data restoration with deep learning

Classical techniques for imputing lost values typically revolve around using a local or global mean wherever a missing value occurs (*Donders et al., 2006*; *Schmitt, Mandel & Guedj, 2015*). When working with discrete data, instead of means, class values that occur with high frequency locally or globally are often used in techniques like hot-deck imputation (*Andridge & Little, 2010*; *Myers, 2011*; *Aljuaid & Sasi, 2016*), k-nearest neighbours (*Cover & Hart, 1967*), decision trees (*Safavian & Landgrebe, 1991*), etc. Other, more complex, techniques work only with multivariate, and primarily continuous, data (*Raghunathan et al., 2001*; *Buuren & Groothuis-Oudshoorn, 2010*). However, all of these techniques fail to properly address the problem of imputing missing discrete data in sequences (*Schafer & Graham, 2002*).

Interestingly, while deep learning has been repeatedly shown to be effective at sequential modelling, there has been little work in applying it to data restoration. Of the methods that do address data restoration, many are designed specifically for the restoration of continuous data (*Duan et al., 2014*; *Zhou & Huang, 2017*; *Niklaus, Mai & Liu, 2017*). Some methods are intended for use only with image data through techniques like in-painting (*Xie, Xu & Chen, 2012*; *Lehtinen et al., 2018*; *Altinel, Ozay & Okatani, 2018*). More recently, Generative Adversarial Networks (GANs) have been used for super-resolution and denoising of images,

although their primary use case is typically image generation (*Goodfellow et al., 2014*; *Ledig et al., 2017*). There are also a number of data restoration methods that assume that all the data is available at once (*Blend & Marwala, 2008*; *Leke, Marwala & Paul, 2015*; *Gondara & Wang, 2017*; *Beaulieu-Jones & Moore, 2017*). However, in the context of data restoration specifically for discrete, streaming data it was only recently demonstrated that a simple LSTM model can be used to restore missing message IDs in automotive data (*Sucholutsky et al., 2019*).

## RESTORER

### Attention

In an RNN, sequences are represented within their hidden states so the output of each hidden state needs to be computed before the model can access the next step in the sequence. This results in a large bottleneck since a number of sequential operations have to be performed that cannot be parallelized. In addition, this results in long path lengths that elements of the sequence have to traverse, which has been shown to be undesirable behaviour when trying to model long-term dependencies (*Hochreiter, Bengio & Frasconi, 2001*). Meanwhile, not only can convolutional layers be computationally expensive (*Chollet, 2017*), but they also need to be stacked in order to connect all inputs to all outputs (*Kalchbrenner et al., 2016*), which in turn increases path length.

Both recurrent and convolutional layers can instead be replaced with self-attention layers as described in the Transformer model in *Vaswani et al. (2017)*. This removes the bottleneck of sequential operations and reduces path length as dot-product attention provides access to the entire history at once. As a result, training time should be decreased, while the ability to learn long-term dependencies increased. We confirm that this result holds in practice by demonstrating it empirically in 'Experiments'. This improvement does come at a cost: an increase in complexity relative to sequence length since the entire sequence is operated on at once and every element attends to every other input. To avoid this new bottleneck, a version of Transformer can be used where input is restricted to neighbourhoods of size $r$ instead of operating on the entire sequence at once (*Vaswani et al., 2017*). Table 1 summarizes the theoretical differences between these layer types across three key metrics.

### Architecture

Restorer is not an RNN or LSTM; it instead follows the Transformer architecture illustrated in Fig. 1 but with modified hyper-parameters as seen in Table 2. For comparison, the LSTM architecture is pictured in Fig. 2. For the various Restorer model versions, $N_{blocks}$ is the number of blocks in the encoder and decoder, $d_{model}$ is the dimension of the embedding and consequently is used as the scaling factor in our scaled dot product attention (instead of $d_k$ as described in *Vaswani et al. (2017)*), $d_k$ is the dimension that keys in attention are projected to, $d_v$ is the dimension that values in attention are projected to, $d_h$ is the size of the hidden layers, and $n_{head}$ is the number of parallel heads in each multi-head attention layer. For the LSTM model, $n_{hid}$ is the number of non-LSTM hidden layers, $n_{lstm}$ is the

**Table 1** Comparison of different layers where $n$ is sequence length, $d$ is dimension of representation, k is kernel size, and r is neighbourhood size (*Vaswani et al., 2017*).

| Layer type | Complexity per layer | Sequential Operations | Maximum Path length |
|---|---|---|---|
| Self-Attention | $O(n^2 \cdot d)$ | $O(1)$ | $O(1)$ |
| Recurrent | $O(n \cdot d^2)$ | $O(n)$ | $O(n)$ |
| Convolutional | $O(k \cdot n \cdot d^2)$ | $O(1)$ | $O(log_k(n))$ |
| Self-Attention (restricted) | $O(r \cdot n \cdot d)$ | $O(1)$ | $O(n/r)$ |

number of LSTM layers, $d_h$ is the size of the non-LSTM hidden layers, $d_{lstm}$ is the size of LSTM layers, and $n_{steps}$ is the number of steps for which the LSTM is unrolled.

As mentioned above, one of the theoretical advantages of the proposed approach, is a large reduction in the number of trainable parameters. In fact, we show empirically that our Restorer architectures have as many as two orders of magnitude fewer parameters than the LSTM model as seen in Table 2.

*Sucholutsky et al. (2019)* implemented the LSTM restoration model in Tensorflow. We have re-implemented it in Keras in order to be more consistent as we implement our Restorer model in Keras based on the implementation of Transformer by *Lsdefine (2018)*.

## EXPERIMENTS

### Data

For consistency during comparison, we utilized a similar dataset to the one that the LSTM restoration model was trained on (*Sucholutsky et al., 2019*). We collected automotive Controller Area Network (CAN) traces from a Lexus RX450 h hybrid SUV. A CAN trace is a long sequence of timestamped messages that consist of an ID and a payload of several bytes of data. We strip away the payload and timestamps and keep only the message IDs in order to obtain a dataset consisting of discrete, sequential data. In particular, 20 traces were collected of the same maneuver, the vehicle decelerating from 20km/h to 0km/h. The train/test split was kept at 75%/25% to be consistent with the experiments in *Sucholutsky et al. (2019)*. Each trace contains a sequence of about 6500 messages made up of 43 unique message IDs. In other words, we are presented with a sequence modelling task with a dictionary size of 43. Transformer-based models have already been shown to outperform LSTMs on classical sequence modelling tasks like language (*Radford et al., 2018*; *Radford et al., 2019*), music (*Huang et al., 2018*), and protein sequences (*Rives et al., 2019*). The CAN dataset is instead representative of alternative tasks with datasets that have relatively few training samples but require very high accuracy for results to be useful.

### Benchmark

To better interpret the results of our experiments, we compare them against the same benchmark as *Sucholutsky et al. (2019)*. This benchmark can be described as either a fast Markov model or a history search, and in essense is a conditional probability maximization method. When there are constraints on data or computational power, Hidden Markov Models can match the performance of LSTMs (*Panzner & Cimiano, 2016*) so we believe

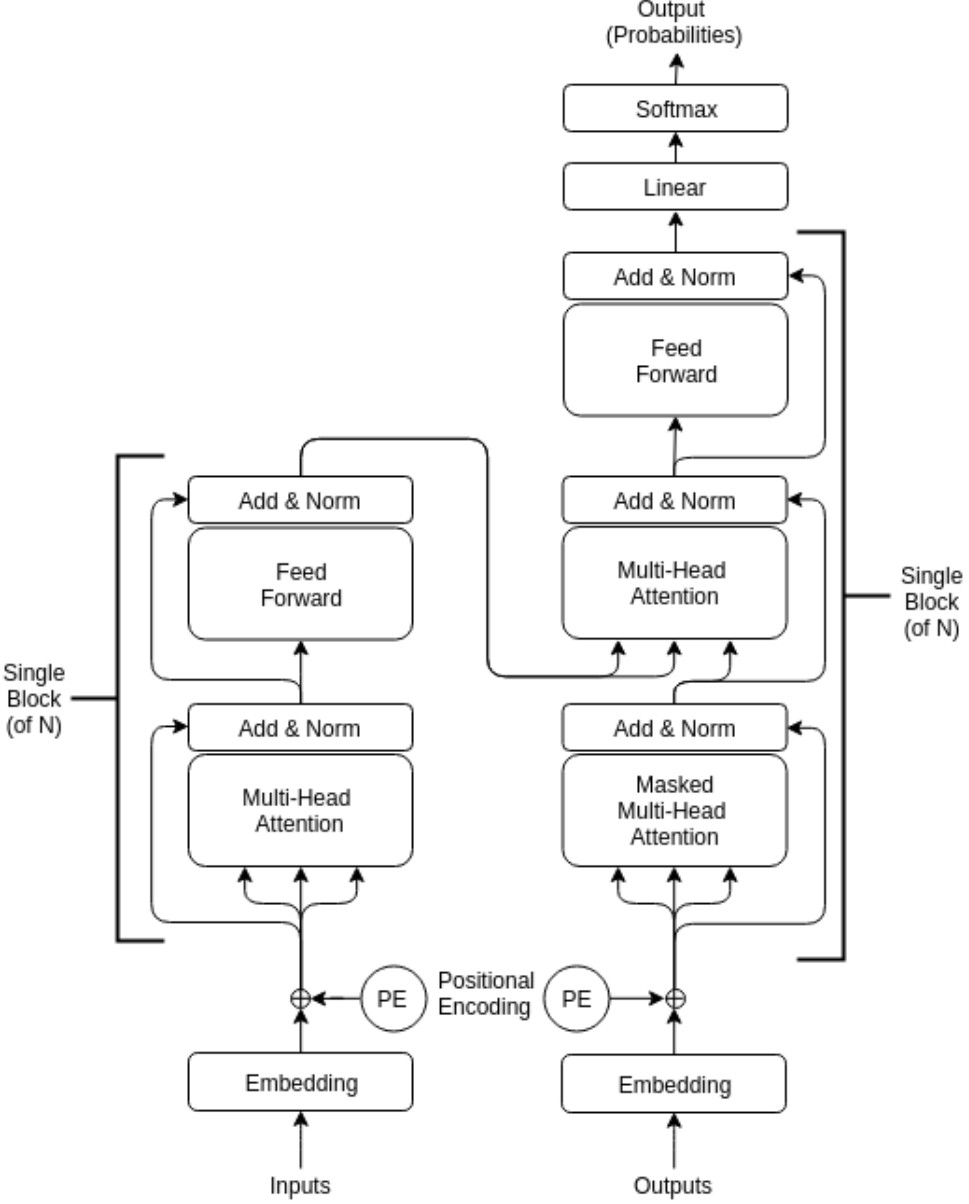

**Figure 1** **Transformer Model Architecture** (*Vaswani et al., 2017*)**:** The Transformer consists of an encoder and decoder each made up of N blocks. Input is a sequence of events, output is a sequence of predicted events.

this is a good benchmark that both the LSTM from *Sucholutsky et al. (2019)* and Restorer can be compared against.

*Sucholutsky et al. (2019)* describes the benchmark in detail, but in short, we select all possible n-grams (*Schonlau, Guenther & Sucholutsky, 2017*; *Lin et al., 2012*; *Lesher, Moulton & Higginbotham, 1999*) from the training data and count the frequency with which they are followed by each event type. When we see an n-gram in the testing data followed by a missing event, we fill in the blank with the most frequently occuring event type

**Table 2  Size comparison of different model versions in terms of number of parameters.**

| Model | | | | | | | params |
|---|---|---|---|---|---|---|---|
| LSTM | $n_{hid}$ | $n_{lstm}$ | $d_h$ | $d_{lstm}$ | $n_{steps}$ | | |
| | 3 | 2 | 88 | 352 | 40 | | 1,640,892 |
| Restorer | $N_{blocks}$ | $d_{model}$ | $d_k$ | $d_v$ | $d_h$ | $n_{head}$ | |
| (main) | 1 | 15 | 64 | 64 | 512 | 4 | 80,824 |
| | 2 | 15 | 64 | 64 | 512 | 4 | 158,873 |
| | 3 | 15 | 64 | 64 | 512 | 4 | 236,922 |
| | 5 | 15 | 64 | 64 | 512 | 4 | 393,020 |
| | 10 | 15 | 64 | 64 | 512 | 4 | 783,265 |
| | 15 | 15 | 64 | 64 | 512 | 4 | 1,173,510 |
| | 20 | 15 | 64 | 64 | 512 | 4 | 1,563,755 |
| (1-head) | 1 | 15 | 64 | 64 | 512 | 1 | 46,264 |
| | 2 | 15 | 64 | 64 | 512 | 1 | 89,753 |
| | 3 | 15 | 64 | 64 | 512 | 1 | 133,242 |
| (mini) | 1 | 15 | 32 | 32 | 88 | 1 | 14,216 |
| | 2 | 15 | 32 | 32 | 88 | 1 | 25,657 |
| | 3 | 15 | 32 | 32 | 88 | 1 | 37,098 |
| (mini2) | 1 | 32 | 16 | 16 | 32 | 1 | 16,704 |
| | 2 | 32 | 16 | 16 | 32 | 1 | 27,488 |
| | 3 | 32 | 16 | 16 | 32 | 1 | 38,272 |
| (micro) | 1 | 16 | 8 | 8 | 16 | 1 | 5,792 |
| | 2 | 16 | 8 | 8 | 16 | 1 | 8,624 |
| | 3 | 16 | 8 | 8 | 16 | 1 | 11,456 |

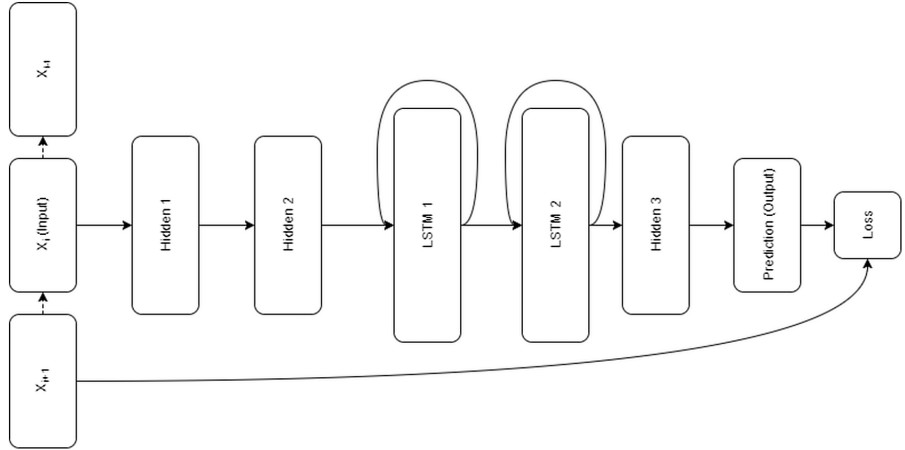

**Figure 2  LSTM Model Architecture (*Sucholutsky et al., 2019*):** The LSTM model consists of two hidden layers followed by two recurrent LSTM layers and one additional hidden layer. Input is a sequence of events, output is a prediction of next event in the sequence. Loss is calculated as a logloss function comparing the true next event to the predicted one.

corresponding to that n-gram. The iterative procedure from *Sucholutsky et al. (2019)* for collecting the frequencies is recreated in Algorithm 1.

**Result:** Dictionary $D$ containing all states and transition frequencies found in the training data

$i = 0$;
**while** $i + n < length(train\_data)$ **do**
    $K = train\_data[i : i + n]$;
    **if** $K \notin D.keys()$ **then**
        Let $S_K$ be a new dictionary;
        **for** *each $m_i$ in the set of the d unique messages* **do**
            $S_K[m_i] := 0$;
        **end**
        $D[K] := S_K$;
    **end**
    $m^* = train\_data[i + n]$;
    $S_K[m^*] = S_K[m^*] + 1$;
    $i = i + 1$;
**end**

**Algorithm 1:** Iteratively learn transition frequencies

Accuracy for this benchmark reaches its peak next-event prediction accuracy of 76.55% (30.07% when reconstructing 40 events forward, 21.1% for 100 events forward) and stays constant for $n \geq 30$, so for consistency with the experiments from *Sucholutsky et al. (2019)* we also use $n = 40$. Tables 3 and 4 which are described in the Results section offer some more benchmark results and some comparisons in accuracy between Restorer, LSTM, and this benchmark model.

## Results

The models are trained using Adam optimizer (*Kingma & Ba, 2014*) with a fixed learning rate for 30 epochs on a randomly selected trace, before moving on to another randomly selected trace and repeating the procedure. Restorer is the restricted neighbourhood version of Transformer that was briefly mentioned as a future direction in *Vaswani et al. (2017)*, so input sequence length can be limited to a neighbourhood of size 40 to be consistent with the LSTM model. As a result, each model received 40 consecutive messages as input and was asked to predict the next 40 messages as output. We trained both models using this training protocol, described in *Sucholutsky et al. (2019)*, to compare the accuracy of the Restorer to that of the LSTM model in a consistent way. One important change when using the restricted version of Transformer is that the model no longer needs to learn to use an end of sequence token as length of the output is predetermined.

Table 3 illustrates that the majority of the different Restorer combinations outperform the LSTM model by several percent, and the benchmark by much more, when using the 40 previous events to predict the next 40. Interestingly, the best performance seems to be

**Table 3  Maximum percent accuracy after n epochs when trained with input and output lengths of 40.**

| Model | | | | | | | 30-ep. acc | 300-ep. acc | 3000-ep. acc |
|---|---|---|---|---|---|---|---|---|---|
| Benchmark | | | | | | | 30.07 | 30.07 | 30.07 |
| LSTM | $n_{hid}$ | $n_{lstm}$ | $d_h$ | $d_{lstm}$ | $n_{steps}$ | | | | |
| | 3 | 2 | 88 | 352 | 40 | | 67.32 | 86.72 | 88.59 |
| Restorer | $N_{blocks}$ | $d_{model}$ | $d_k$ | $d_v$ | $d_h$ | $n_{head}$ | | | |
| (main) | 1 | 15 | 64 | 64 | 512 | 4 | 62.88 | 90.01 | 92.56 |
| | 2 | 15 | 64 | 64 | 512 | 4 | 74.76 | 88.12 | 93.41 |
| | 3 | 15 | 64 | 64 | 512 | 4 | 74.19 | 89.56 | **94.00** |
| | 5 | 15 | 64 | 64 | 512 | 4 | 74.17 | 83.01 | 84.44 |
| | 10 | 15 | 64 | 64 | 512 | 4 | 9.46 | 9.47 | – |
| (1-head) | 1 | 15 | 64 | 64 | 512 | 1 | 69.40 | 88.22 | 90.04 |
| | 2 | 15 | 64 | 64 | 512 | 1 | **74.79** | 89.87 | 92.71 |
| | 3 | 15 | 64 | 64 | 512 | 1 | 72.01 | 90.00 | 92.83 |
| (mini) | 1 | 15 | 32 | 32 | 88 | 1 | 68.91 | 87.01 | 88.85 |
| | 2 | 15 | 32 | 32 | 88 | 1 | 71.67 | 85.83 | 91.49 |
| | 3 | 15 | 32 | 32 | 88 | 1 | 71.66 | 89.85 | 93.15 |
| (mini2) | 1 | 32 | 16 | 16 | 32 | 1 | 71.66 | 91.02 | 93.28 |
| | 2 | 32 | 16 | 16 | 32 | 1 | 73.74 | 90.89 | 94.01 |
| | 3 | 32 | 16 | 16 | 32 | 1 | 74.60 | **91.27** | 93.69 |
| (micro) | 1 | 16 | 8 | 8 | 16 | 1 | 59.09 | 84.80 | 86.45 |
| | 2 | 16 | 8 | 8 | 16 | 1 | 64.87 | 85.35 | 90.33 |
| | 3 | 16 | 8 | 8 | 16 | 1 | 64.29 | 88.96 | 92.22 |

**Table 4  Percent accuracy on different output lengths after training using all data at once for 3,000 epochs with input length of 40 and a new target output length randomly selected every 30 epochs.**

| Model | | 1 out acc | 10 out acc | 20 out acc | 40 out acc | 60 out acc | 80 out acc | 100 out acc |
|---|---|---|---|---|---|---|---|---|
| Benchmark | | 76.55 | 45.41 | 37.01 | 30.07 | 25.70 | 22.99 | 21.10 |
| Restorer | $N_{blocks}$ | | | | | | | |
| (main) | 1 | 91.22 | 92.09 | 92.23 | 92.16 | 75.21 | 56.41 | 45.13 |
| | 2 | **91.98** | **92.49** | **92.54** | 92.44 | 78.65 | 58.98 | 47.18 |
| | 3 | 91.89 | 92.37 | 92.44 | **92.58** | **92.56** | **92.56** | **88.41** |
| | 5 | 91.66 | 91.99 | 91.94 | 92.08 | 92.21 | 92.31 | 86.96 |

achieved by smaller architectures with either two or three identical blocks, although the smallest architecture, micro, has a noticeable drop in accuracy. The larger architectures seem to primarily suffer from overfitting, as training accuracy quickly begins to exceed validation accuracy during their training. Models with 10 or more blocks did not converge at all past 10% test accuracy, likely due to the fixed learning rate being used. Figure 3 shows the full testing accuracy for the top configuration of each Restorer model version. While the increase in testing accuracy certainly slows down by 3000 epochs, we found that even after 6000 epochs small increases were still being made.

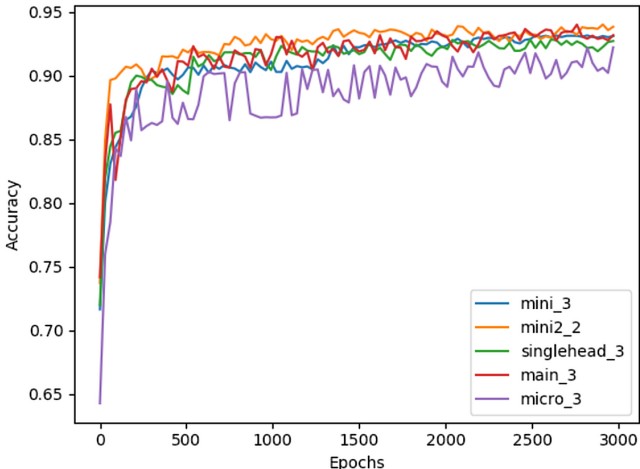

Figure 3  Testing accuracy of best Restorer configurations measured every 30 epochs of training.
Model titles in the legend follow the format [model name]_[# of blocks].

Table 5  Training time in seconds on single V100 GPU.

| Model | 30-ep. time | 300-ep. time |
| --- | --- | --- |
| LSTM | 77.64 | 807.25 |
| Restorer (main, $N_{blocks} = 3$) | 36.22 | 231.95 |

Table 6  Maximum percent accuracy after n epochs when trained with input and output lengths of 40 and using all training data at once.

| Model | $N_{blocks}$ | $d_{model}$ | $d_k$ | $d_v$ | $d_h$ | $n_{head}$ | 30-ep. acc | 300-ep. acc | 3000-ep. acc |
| --- | --- | --- | --- | --- | --- | --- | --- | --- | --- |
| Restorer (main) | 1 | 15 | 64 | 64 | 512 | 4 | 93.72 | 94.69 | 94.79 |
| | 2 | 15 | 64 | 64 | 512 | 4 | **94.16** | **95.06** | **95.10** |
| | 3 | 15 | 64 | 64 | 512 | 4 | 91.48 | 94.72 | 94.82 |
| | 5 | 15 | 64 | 64 | 512 | 4 | 84.36 | 84.43 | 93.11 |

On an Amazon Web Services (AWS) p3.2xlarge machine with a single V100 GPU and a batch size of 512, we found that the Restorer main model with 3 blocks was on average 2–4 times as fast as the LSTM model, as shown in Table 5.

While this method of training by selecting one random trace at a time does achieve high accuracy and lead to faster training times, we found that by concatenating all of the training data and training Restorer on it all at once, higher accuracy can be achieved. Table 6 shows that the Restorer achieves one to two percent higher accuracy when trained in this way instead of the trace-by-trace method.

We also found that when Restorer was trained exclusively with output lengths of 40, it generalized poorly to test cases with longer or shorter output sequences. In order to train models that would generalize to other output lengths, we found that changing the output length every 30 epochs to a random integer between 1 and 100 was an effective method. Table 4 summarizes the average accuracy achieved by Restorer models when tested with different output lengths. In this scenario, models with more blocks generalized better than those with fewer. It appears that when the length of the output sequence is fixed and known in advance, smaller models should be used, but when this length is either unknown or variable, it is better to use larger models and train in this stochastic way.

## Case study: timed regular expression mining on CAN traces

It is important to verify that the quality of restoration is not just high in terms of accuracy, but also in terms of the effect on end-stage algorithms that would make use of the restored data. For consistency, we perform a case study using the same complex downstream algorithm as in *Sucholutsky et al. (2019)*.

Time of occurrence of an event plays a key role in the domain of real-time systems (*Lamport, 1978*; *Dwyer, Avrunin & Corbett, 1999*). Recently, there has been tremendous activity in reverse engineering of complex real-time systems by mining temporal properties, typically by making use of template-based mining frameworks, that reflect the common behavior of these systems (*Lemieux, Park & Beschastnikh, 2015*). Since most programs in industry and elsewhere lack formal specifications, mined specifications play a key role in the life of software as they can be used for tasks like testing (*Dallmeier et al., 2010*) or verification (*Kincaid & Podelski, 2015*).

State-machine based approaches have become quite popular in the domain of both active and passive learning of system specifications. The behavior of a system is learned in the form of a state machine that can be employed for numerous tasks such as run-time monitoring, run-time verification, debugging, etc. In the context of real-time systems, the focus of these temporal specifications is inclined towards a quantitative (actual duration of time between events) notion of time rather than qualitative (ordering of events) notion of time. For example, the response of an interrupt handler should be bound within its predefined time constraints. These time constraints are critical to the safe operation of real-time systems since a delayed response can lead to a faulty system operation.

System traces used for mining temporal specifications for a system are quite often lossy due to various software and hardware issues such as a buffer overflow or a memory leak. Therefore, recovery of data/traces will play a very important role in not only improving understanding of system behavior but also performing better fault diagnosis. Mining of Timed Regular Expressions (TREs) (*Eugene Asarin, 2002*) was proposed by *Cutulenco et al. (2016)*; *Narayan et al. (2018)*. The method proposed by *Eugene Asarin (2002)* to synthesize a timed automaton for a given TRE is the basis of the proposed approach. The timed automaton is then used as a checker to verify whether traces satisfy the corresponding TRE.

The mining algorithm is executed on three types of traces for the purpose of evaluation: (a) Normal traces, (b) Lossy traces, and (c) Restored traces (*Sucholutsky et al., 2019*). The Normal traces in this case correspond to the original traces obtained from a real vehicle as

**Table 7 Percentage deviation in total number of mined TRE instances in restored traces and lossy traces at each level of loss when compared to the number mined in normal traces.**

| Trace type | Loss = 5% | Loss = 10% | Loss = 15% | Loss = 20% | Loss = 25% |
|---|---|---|---|---|---|
| Lossy traces | 6.2% | 17.8% | 21.6% | 33.37% | 53.99% |
| Restored traces | 1.16% | 4.6% | 6.55% | 7.96% | 9.53% |

described in 'Experiments'. We do not have a large corpus of lossy CAN traces so instead lossy traces were generated from Normal traces by introducing random loss of events. We introduced loss of events at five levels: 5%, 10%, 15%, 20%, and 25%. Higher levels of loss are extremely unlikely to occur in a real vehicle aside from total failure of multiple components and as such are not included; nonetheless, exploratory experiments suggested that final results for higher levels of simulated loss were consistent with the pattern seen among these five selected levels. All losses were random to ensure that we are able to approximate a real-world scenario, as bugs, sensor defects, buffer overflows, cyberattacks, disconnects, and other similar events can all cause loss at any point within a trace. Lastly, the Restored traces were obtained from the Restorer main model with $N_{blocks} = 3$ that was trained on the Normal traces. We ensure that timestamps remain in order with the Normal traces since both time and order of occurrence of events is of key importance for specification in the form of TREs. Two TRE templates from *Narayan et al. (2018)* are used with a time interval parameter of 0 to 1,000 as was done in *Sucholutsky et al. (2019)*.

**T-1(response):** $(\langle P \rangle^* . (\langle P.\langle S \rangle^* . S \rangle [0, 1000]) . \langle P \rangle^*) +$

**T-2(alternating):** $(\langle P|S \rangle^* . (\langle P.\langle P|S \rangle^* . S.\langle P|S \rangle^* \rangle [0, 1000])) +$

We evaluated the performance of the Restorer main model with $N_{blocks} = 3$ by comparing the percent of TRE instances mined by the framework of *Narayan et al. (2018)*. In Table 7, we present the percent of instances that were not found in the lossy and restored traces when compared with normal traces. Low deviation values are desirable as deviation is indicative of information loss. It is evident from the results in Table 7 that the quality of the Restorer restoration is extremely high. The Restorer is able to reduce the deviation in number of mined instances from 6.2% to 1.16% in case of low levels of loss and there is a remarkable reduction of almost 45% in the deviation of number of instances mined in cases with high levels of loss.

## CONCLUSION

Restoring lost data is an important intermediate problem in machine learning as many algorithms rely on clean data for input. Previous solutions relied on LSTMs to perform the restoration of discrete data in sequences. We have demonstrated that models making use of attention mechanisms based on the recently proposed Transformer model are faster, smaller, and achieve better accuracy than the LSTM restoration model. We have shown that the exact architecture can be tailored to the task at hand based on the associated constraints. For example, for compute-constrained or time-constrained tasks, a smaller architecture can be chosen to minimize training or inference times and number of parameters. Meanwhile,

for tasks where the length of the output sequence is variable or unknown beforehand, a larger architecture with more blocks achieves better accuracy.

There are several limitations of this study that merit discussion. First off, we avoided using bidirectional methods because they would have a large impact on the latency of algorithms running downstream from Restorer when working with streams of data. However, in order to improve accuracy, it may be useful to develop a mechanism for using newly processed steps of the input sequence to retroactively correct predictions made by Restorer. Second, while we have tested Restorer on real data, it will be important to conduct additional studies with other datasets to further establish Restorer as an effective solution for data restoration. Finally, we have shown that Transformer-based models can be used for even more sequence modelling tasks than the already wide-range described in *Vaswani et al. (2017)*; however, more experiments with a larger variety of tasks may need to be conducted before Transformer-based models are fully accepted as a state-of-the-art framework for general sequence modelling rather than just natural language modelling. That being said, the results from the CAN data experiments and the case study are certainly promising.

In general, we have shown that Restorer can achieve accuracies of up to 95% when restoring long sequences of missing CAN data, beating out the benchmark method by a wide margin. When comparing Restorer to LSTM-based methods for restoring CAN data, we demonstrated that using Restorer leads to a reduction in model sizes by up to two orders of magnitude, an up to 4-fold reduction in training times, and an increase of up to 7% in accuracy of data restoration. We additionally demonstrated in a case study that our method successfully improves the performance of complex downstream algorithms.

### Funding
The authors received no funding for this work.

### Competing Interests
The authors declare there are no competing interests.

### Author Contributions
- Ilia Sucholutsky and Apurva Narayan conceived and designed the experiments, performed the experiments, analyzed the data, contributed reagents/materials/analysis tools, prepared figures and/or tables, performed the computation work, authored or reviewed drafts of the paper, approved the final draft.
- Matthias Schonlau conceived and designed the experiments, approved the final draft, supervision.
- Sebastian Fischmeister conceived and designed the experiments, approved the final draft, data acquisition.

### Data Availability
The raw data and code are available in the Supplemental Files. The system under test (SUT) is a Lexus RX450h hybrid SUV owned by WatCAR. The data was collected by Sunaal Mathew, Jeff Graansma, Kevin Cochran at the end of April 2015. CAN data was collected using Vector CANoe hardware. There are no known anomalies in the dataset. Twenty traces were collected of the same maneuver, the vehicle decelerating from 20 km/h to 0 km/h. Each of the 20 Supplemental Files within Data S1 correspond to the PGN column extracted from a trace file.

## Supplemental Information

Supplemental information for this article can be found online at http://dx.doi.org/10.7717/peerj-cs.210#supplemental-information.

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
