# Peer review of "Pay attention and you won’t lose it: a deep learning approach to sequence imputation"

_PeerJ Computer Science, doi:10.7717/peerj-cs.210_

## Round 0.1 · original submission · Major Revisions

One of the reviewers commented on the additional contributions this research presents compared to your previous publication at IJCNN. The other reviewer, who provided very detailed comments, raised the question of how the presented algorithm apply to other datasets.

Reviewer 1 ·

Basic reporting

no comment

Experimental design

no comment

Validity of the findings

no comment

Additional comments

In this paper, the authors proposed the use of attention mechanisms for the imputation of values in discrete sequences using LSTMs. According to the authors, this is the first time that this type of imputation mechanisms are used in discrete sequences and they argue this fact.
They use the architecture of Vaswani's transformer model and test the results in the CAN data set.

The paper is well written and well understood. The results compared to an LSTM baseline are surprising in terms of accuracy and time reduction. However, I am not clear about the novelty of the idea with respect to an accepted paper written by the same authors in IJCNN this year.

I'm sure there are differences, beyond the implementation of the model in Keras instead of TensorFlow, but I think that the authors should record each and every one of the novelties that this new study represents with respect to the previous publication. I am referring to the differences in the mechanism of attention and architecture, as well as the extension of experimental results and tests that are carried out in this paper. A new section (or subsection) should be included for this purpose.

From my point of view, with a clear justification of the novelty, the paper could be accepted, as it represents an interesting advance in the pre-processing of discrete sequence data.

·

Basic reporting

Line 40: I’d still delete “only” from here so it reads “very few attempts” rather than “only very few attempts”.

I’d prefer a legend on Figure 2 that didn’t rely on underscores and clarified the names a little more, particularly since these names are not defined in the manuscript (I appreciate that the configuration names are clear enough in intent, but they could definitely be improved).

Line 181: I don’t think this comma is needed (or another matching comma is missing from earlier in the sentence).

Lines 206–207: Perhaps “In the context of…” (adding “the”) here and a comma after “systems”.

Line 248: I don’t think the first comma (after “model”) is needed (or another matching comma is missing from earlier in the sentence).

Experimental design

No comment.

Validity of the findings

No comment.

Additional comments

The revised version of the manuscript addresses my primary concern from last time (the reliance on an anonymous unpublished manuscript) and addresses most of the other comments I made also. I have only a few minor comments above, one which is a repeat from the previous version of the manuscript.

Reviewer 3 ·

Basic reporting

This paper studies the use of LSTMs for data restoration. The current state-of-the-art performs a large number of sequential operations, slowing the training down. The aim of this paper is to alleviate such issues by means of attention mechanisms. This is inspired in the paper of Vaswani 2017, and the authors use a version of ‘Transformer with modified hyper-parameters. As such, the novelty is not extremely high, but I believe that it is definitely of interest for the deep learning community.

Overall, the paper is well-written and is easy to follow. I believe, however, it requires to extend a few sections to contextualise better their proposal.

Experimental design

- The authors are using data collected from a CAN of an SUV. Following the description in section 3.1, it is unclear what kind of machine learning problem the authors are tackling. I don’t understand the meaning of 43 unique messages Ids. The authors need to clarify what is going to be input into the proposed model.

- The training/test split seems to be a bit too simplistic. Why should you apply a fold-cross validation?

- Why table 4 doesn’t report the time for 3000 epochs?

Validity of the findings

The algorithm seems to be quite competitive in comparison with the previous authors’ proposals. However, this is only applied on a single dataset; the authors should apply this on more datasets to check if the proposed method actually restores well in other scenarios.

Additional comments

The introduction is missing general references on data preprocessing, data cleaning, obtaining ‘smart data’ from big data. I would also extend the introduction to motivate better the problem of restoration and when it is applied.

I very much like the discussion about Deep learning not being used for data-preprocessing or cleaning. I wonder if the authors could elaborate a bit more on that regard. For example, data augmentation, GANs could be considered preprocessing as well (as oversampling technique). But it is definitely true that it is frequently not being used for data filtering or noise detection. I think that would also be a good opportunity to contextualise better the proposed method.

Is restoration a missing values imputation mechanism? If yes, it is unclear in the introduction if the proposed method is devised to tackling images/videos or ‘any kind’ of data. In the case of deep learning, the authors should also mention in-painting techniques as related work, and for standard classification/regression/time series problems, there are many missing values imputation mechanisms which are related. I slightly disagree that the classical techniques for missing values imputation revolve around local and global means (they are of course the basic), but classical techniques also include techniques such as the k-nearest neighbours.

The description of the proposed ‘restorer’ method is a bit shallow. The authors have assumed lots of background information which could be solved by adding some additional information in the background. Figure 1 should belong to section 2.2.

---

## Round 0.2 · accepted · Accept

Two of the reviewers are satisfied with the revision. I am happy to accept the revised manuscript.

Reviewer 1 ·

Basic reporting

The authors have satisfactorily addressed all my comments and now the paper can be accepted for publication.

Experimental design

The authors have satisfactorily addressed all my comments and now the paper can be accepted for publication.

Validity of the findings

The authors have satisfactorily addressed all my comments and now the paper can be accepted for publication.

Additional comments

The authors have satisfactorily addressed all my comments and now the paper can be accepted for publication.